# Rare β-Resorcylic Acid Derivatives from a Halophyte-Associated Fungus *Colletotrichum gloeosporioides* JS0419 and Their Antifungal Activities

**DOI:** 10.3390/md20030195

**Published:** 2022-03-07

**Authors:** Sunghee Bang, Jaekyeong Kim, Jiwon Oh, Ji-Seok Kim, Seong-Ryong Yu, Stephen Deyrup, Yong-Sun Bahn, Sang Hee Shim

**Affiliations:** 1College of Pharmacy, Duksung Women’s University, Seoul 01369, Korea; scbsh4331@hanmail.net (S.B.); 5j1ohoh@gmail.com (J.O.); 2Natural Products Research Institute, College of Pharmacy, Seoul National University, Seoul 08826, Korea; worud0435@snu.ac.kr; 3Department of Biotechnology, College of Life Science and Biotechnology, Yonsei University, Seoul 03722, Korea; wltjrs123@naver.com (J.-S.K.); dragonyoo@naver.com (S.-R.Y.); ysbahn@yonsei.ac.kr (Y.-S.B.); 4Department of Chemistry and Biochemistry, Siena College, Londonville, NY 12211, USA; sdeyrup@siena.edu

**Keywords:** *Colletotrichum gloeosporioides*, polyketides, β-resorcylic acid lactones (RALs), non-esterification, antifungal

## Abstract

Six new β-resorcylic acid derivatives (**1**–**5** and **7**) were isolated from a halophyte-associated fungus, *Colletotrichum gloeosporioides* JS0419, together with four previously reported β-resorcylic acid lactones (RALs). The relative and absolute stereochemistry of **1** was completely established by a combination of spectroscopic data and chemical reactions. The structures of the isolated compounds were elucidated by analysis of HRMS and NMR data. Notably, compounds **1**–**3** had a β-resorcylic acid harboring a long unesterified aliphatic side chain, whereas the long aliphatic chains were esterified to form macrolactones in **4**–**9**. Among the isolated compounds, monocillin I and radicicol showed potent antifungal activities against *Cryptococcus neoformans*, comparable to clinically available antifungal agents and radicicol showed weak antifungal activity against *Candida albicans*. These findings provide insight into the chemical diversity of fungal RAL-type compounds and their pharmacological potential.

## 1. Introduction

A variety of bioactive metabolites have been reported from plant-associated microorganisms [1]. Notably, the microorganisms have been reported to show mutualistic symbiosis with halophytes to help them survive in high saline conditions [2]. The discovery of bioactive secondary metabolites and the elucidation of relationship(s) between microbes and the host halophyte have become an attractive research aim, owing to their functional diversity [3].

One of the major polyketides produced by fungi is resorcylic acid lactones (RALs), which constitute a β-resorcyclic acid unit (2,4-dihydroxy-benzoic acid) embedded in a 14-membered lactone [4]. Since the first naturally occurring RAL, radicicol, was isolated from *Monocillium nordinii* in 1953 and found to be an excellent Hsp20 inhibitor [2], other RAL-type compounds, such as LL-Z1640-2, hypothemycin, monocillins (I–V), zeaenol, aigialomycins (A–E), pochonins (A–P), paecilomycins (A–F), and cochliomycins, have been subsequently reported to date [5,6,7,8,9,10,11,12,13,14]. All of these RAL-type compounds exhibit diverse biological activities, such as antifungal, cytotoxic, antimalarial, antiviral, antiparasitic, nematicidal, estrogenic, protein tyrosine kinase, and ATPase inhibition, resulting in significantly increasing attention in the field of natural product chemistry [10,11,15,16,17,18,19,20].

As part of our efforts to discover new bioactive compounds from halophyte-associated fungi, we investigated the chemical profiles of fungal extracts using liquid chromatography coupled with a diode array detector and mass spectrometry (LC-DAD-MS). We selected the fungal strain *C. gloeosporioides* JS0419, isolated from the halophyte *Suaeda japonica*, for further analysis, because it produces a variety of secondary metabolites with unique UV patterns [21,22,23,24]. After fermentation and subsequent purification, we identified two distinct classes of new compounds in *C. gloeosporioides* JS0419. The first class comprised glycosylated cyclic lipodepsipeptides (colletotrichamides A–E) with moderate neuroprotective activities against glutamate-induced neurotoxicity in hippocampal HT22 cells, which we recently reported [25]. The other class comprised polyketides represented by β-RALs, which included non-lactonized and lactonized lactones. Chemical investigation of *C. gloeosporioides* JS0419 isolated from the leaves of *S. japonica* led to the isolation of nine polyketides (**1**–**9**). Among them, we identified the previously reported compounds, nordinonediol (**6**) [10], monocillin I (**8**) [26], and radicicol (**9**) [27] by comparing their spectroscopic data with those in previous reports. Here, we report the isolation, structural elucidation, and bioactivity of these polyketide compounds (Figure 1).

## 2. Results

### 2.1. Structure Determination of the Compounds

Compound **1** (colletogloeic acid A) was isolated as a yellowish amorphous solid and analyzed for its molecular formula (C_18_H_26_O_8_) based on (+)HRESIMS (observed [M + Na − H_2_O]^+^ at *m/z* 375.1420, calcd for *m/z* 375.1414), which was supported by ^1^H-NMR, ^13^C-NMR, and heteronuclear single quantum coherence (HSQC) data (Table 1). The ^1^H NMR spectrum of **1** displayed one olefinic proton (*δ*_H_ 6.31), two meta-coupled aromatic protons (*δ*_H_ 6.29 (d, *J* = 2.0 Hz) and 6.28 (d, *J* = 2.0 Hz)), three carbinol protons (*δ*_H_ 4.00, 3.66, and 3.42), ten aliphatic protons (*δ*_H_ 2.52-1.42), and one methyl proton (*δ*_H_ 1.20). The ^13^C NMR spectrum together with the HSQC data of **1** revealed eighteen carbons, one carboxylic acid carbon (*δ*_C_ 168.1); eight sp^2^ carbons (*δ*_C_ 167.7~99.6), three of which were protonated; three oxygenated sp^3^ methine carbons (*δ*_C_ 76.1, 72.9, and 65.6), five aliphatic sp^3^ carbons (*δ*_C_ 42.4, 34.1, 33.4, 28.1, and 26.5), and one methyl carbon (*δ*_C_ 24.6). The presence of β-resorcylic acid (2,4-dihydroxybenzoic acid) was determined based on the *meta*-coupled proton signals (*J* = 2.0 Hz) together with two oxygenated sp^2^ carbons and a carboxylic carbon signal. Interpretation of ^1^H-^1^H COSY allowed the assignment of a long aliphatic chain [CH_2_-CH_2_-CH_2_-CH_2_-CH(O)-CH(O)-CH_2_-CH(O)-CH_3_] corresponding to C-10 to C-18, and we confirmed its connection with the β-resorcylic acid moiety according to the HMBC spectrum (Figure 2). Although there are many overlapping signals for CH_2_ in the ^1^H NMR spectra, all the methylene protons and carbons could be assigned by interpreting HMBCs from isolated methylene protons (H_2_-10) and oxymethines (H-14, H-15, and H-17), as shown in Figure 2. HMBCs of an olefinic proton H-8 (*δ*_H_ 6.31) of the aliphatic chain with C-2 (*δ*_C_ 99.6) and C-6 (*δ*_C_ 103.9) of the β-resorcylic acid moiety indicated that the long-chain moiety was attached to C-7. Additionally, we considered the possibility that the long aliphatic chain of **1** was macrolactonylated through an ester bond between C-1 and C-17, as many RAL-type compounds have previously been reported in fungi. However, it was more probable to have a free hydroxyl group at C-17 rather than to be esterified to form a macrolactone ring due to the chemical resonance of H-17 (*δ*_H_ 4.0) in the long chain. Evidently, no HMBC correlation from H-17 to the carbonyl carbon C-1 supported the presence of free hydroxyl groups. Although a number of RAL-type compounds have been reported from fungi, to our knowledge, this is a rare case to isolate β-resorcylic acid with an unesterified long aliphatic side chain. We then confirmed the presence of β-resorcylic acid with a non-esterified aliphatic chain by a simple chemical reaction. Compound **1** was acetylated to confirm the number of exchangeables in the side chain. Acetylation of **1** caused downfield shifts of three oxymethine protons (H-14, H-15, and H-17) in the aliphatic side chain by ~1 ppm (compared with those in compound **1**), thus completing the planar structure of compound **1** (Appendix A). Compound **1** was enolated at C-8 and C-9, whereas most of the previously reported RAL-type compounds have a keto group at C-9. To the best of our knowledge, the only precedent example of RAL with an enol group at C-8 and C-9 is pochonin P [26]. Therefore, the present study represents the second report on the natural occurrence of β-resorcylic acid derivatives with an enol chain.

Compound **2** (colletogloeic acid B) was a yellowish amorphous solid and determined to be chlorinated according to the characteristic isotopic pattern in its MS spectrum. Its molecular formula was found to be C_18_H_25_ClO_9_ based on (+)HRESIMS (observed [M + Na − H_2_O]^+^ at *m/z* 425.1010) and spectroscopic data. This chlorinated compound had one aromatic proton, one olefinic proton, and four oxymethine proton signals. The C-6 of the β-resorcylic acid moiety was confirmed to be chlorinated based on HMBCs of the aromatic proton H-4 (*δ*_H_ 6.42) and the olefinic proton H-8 (*δ*_H_ 6.72) with C-6 (*δ*_C_ 108.3). Detailed analysis of ^1^H-^1^H COSY revealed the presence of a 1,3,4,6-tetraol system in the aliphatic chain instead of the 1,2,4-triol system shown in **1**. The position of the additional carbinol group (*δ*_H_ 3.91; *δ*_C_ 70.4) was confirmed to be C-12 by HMBCs of H_2_-10 (*δ*_H_ 2.75 and 2.67) with C-8 (*δ*_C_ 101.9) and C-12 (*δ*_C_ 70.4), suggesting its planar structure (Figure 2).

Compound **3** (colletogloeic acid C) was a yellowish amorphous solid with the molecular formula C_18_H_24_O_7_ according to (+)HRESIMS (observed [M + Na − H_2_O]^+^ at *m/z* 357.1312, calcd [M + Na − H_2_O]^+^ at *m/z* 357.1309). Although the NMR spectral data of **3** were similar to those of **1**, an additional double bond in *trans* at *δ*_H_ 6.54 (1H, dt, *J* = 15.5, 2.5 Hz) and 6.16 (1H, d, *J* = 15.5 Hz) and the lack of the oxygenated methine group revealed a distinct structural difference between **1** and **3**. The additional double bond was positioned at C-10 and C-11 based on the HMBC correlations of H-10 (*δ*_H_ 6.16) with C-8 (*δ*_C_ 106.0) and C-12 (*δ*_C_ 33.8), as well as H-11 (*δ*_H_ 6.54) with C-9 (*δ*_C_ 153.5) and C-13 (*δ*_C_ 26.1). Additionally, a spin system CH(O)-CH_2_-CH(O)-CH_3_ corresponding to C15(O)-C16-C17(O)-C18 obtained by ^1^H-^1^H COSY indicated hydroxylation at C-15 and C-17, which was also supported by HMBCs.

Compound **4** (Colletoresorcylic lactone) was a yellowish amorphous solid and its molecular formula was found to be C_18_H_24_O_7_ based on the HSQC NMR and (+)HRESIMS (observed [M + Na]^+^ at *m/z* 375.1419). Compound **4** showed different UV profiles from compounds **1**–**3**. While **1**–**3** had a strong UV absorption at 244 nm and a medium absorption at 320 nm, **4** showed three UV absorptions at 211, 262, and 298 nm, suggestive of a lactonized resorcylic acid. The ^1^H, ^13^C, and ^1^H-^1^H COSY NMR data of **4** were similar to those of **1** in the presence of the 2,4-dihydroxybenzoic acid moiety and a 1,2,4-triol group, CH_3_-CH(O)-CH_2_-CH(O)-CH(O)-CH_2_, in the aliphatic chain. A methylene group at *δ*_H_ 4.48 and *δ*_H_ 3.82 and a ketone carbon (*δ*_C_ 212.5) in **4** instead of the enol group indicated that **4** had a keto group. Notably, compound **4** had an oxymethine proton signal at *δ*_H_ 4.95 unlike previous compounds **1**–**3**, suggesting that **4** had an ester group. An HMBC correlation of the oxymethine proton *δ*_H_ 4.95 with the carboxylic carbon (C-1) at *δ*_C_ 171.9 in addition to two oxymethine carbons (C-14 and C-17) indicated the presence of the 12-membered lactone ring. Additionally, HMBCs of the methylene protons H_2_-8 (*δ*_H_ 4.48 and 3.82) with C-2 (*δ*_C_ 107.5), C-6 (*δ*_C_ 113.9), and C-9 (*δ*_C_ 212.5) confirmed the presence of the keto group at C-9.

Compound **5** had the molecular formula C_18_H_23_ClO_7_ according to (+)HRESIMS data and it exhibited a similar UV pattern to **4**, characteristic for RAL-type compounds. Moreover, this was supported by an HMBC of H-17 (*δ*_H_ 5.43) in a long chain with the carboxylic carbon C-1 (*δ*_C_ 172.5). Chlorination at C-6 of the β-resorcylic acid moiety was assigned based on its HMBCs of H-4 and H_2_-8 with C-6 along with typical isotopic mass fragmentation. Compound **5** was structurally similar to the known compound nordinonediol (**6**), except for the presence of chlorine at C-6 of the RAL moiety.

Compound **7** was a yellowish amorphous solid with a molecular formula of C_23_H_28_O_12_ based on (+)HRESIMS data. Additionally, we determined **7** to be a RAL-type compound based on its characteristic UV absorption, which was also supported by NMR resonances for an oxymethine group (*δ*_H_ 5.38; *δ*_C_ 72.3) representative of a lactone. Unlike previous RAL-type compounds **5** and **6**, **7** had a conjugated double bond, which was confirmed by proton resonances and their coupling constants (*δ*_H_ 6.08 (d, *J* = 16.0 Hz, H-10), 7.59 (dd, *J* = 16.0, 10.0 Hz, H-11), 6.22 (t, *J* = 10.0 Hz, H-12), and 5.77 (dd, *J* = 10.0, 4.5 Hz, H-13)) together with ^1^H-^1^H COSY (Table 2). HMBCs of the olefinic protons at *δ*_H_ 6.08 and 7.59 with the ketone at C-9 allowed the connection of the conjugated double bond with the ketone C-9. Two oxymethines that appeared in a relatively higher field (*δ*_H_ 3.36 (m, H-14), *δ*_C_ 56.6 (C-14); 3.08 (dt, *J* = 8.5, 3.0 Hz, H-15), 56.9 (C-15)) than the diol group were indicative of epoxide functionality. Moreover, the degrees of unsaturation suggested by the molecular formula were supportive of the epoxide ring. The conjugated double bond was then connected to the epoxide group according to ^1^H-^1^H COSY and HMBC correlations of the epoxide protons, H-14 and H-15 with C-13. The geometries of the double bonds (C-10/C-11, C-11/C-12, and C-12/C-13) were determined to be *E*, *Z*, and *Z* configurations, respectively, based on their coupling constants. Additionally, ^1^H and ^13^C resonances downfield (*δ*_H_ 5.65, *δ*_C_ 103.4; *δ*_H_ 4.22, *δ*_C_ 88.3; *δ*_H_ 4.18, *δ*_C_ 73.7; *δ*_H_ 4.09, *δ*_C_ 71.1; *δ*_H_ 3.69 and 3.65, *δ*_C_ 63.2) suggested the presence of a pentose, which was confirmed to be ribose by comparing its NMR resonances with those of the references [28]. We confirmed the location of the ribose by an HMBC from the anomeric proton H-1′ (*δ*_H_ 5.65) with C-5 (*δ*_C_ 156.5). The ribose moiety was in an α configuration based on the coupling constant of the anomeric proton (^3^*J*_HH_ = 4.5 Hz) [28].

### 2.2. Establishment of Stereochemistry

Colletogloeic acids A-C (**1**–**3**) have several stereogenic centers in their chain. We employed comprehensive spectroscopic analysis and various chemical reactions to establish their relative and absolute stereochemistries. The relative configurations of the consecutive stereogenic centers in the aliphatic chain were initially determined by measuring the NMR data of their acetonide derivatives. Colletogloeic acid A (**1**) was derivatized with 2,2-dimethoxy propane to obtain the acetonide derivative **1a**. By interpreting HMBC data of **1a** together with 1D NMR, 1,2-diol at C-14 and C-15 was confirmed to be acetonylated. The ^13^C resonances for the two methyl groups of the acetonide derivative, which appeared at *δ*_C_ 29.1 and 26.3, indicated a *syn* configuration of the dihydroxyl groups at C-14 and C-15 [29,30]. At this point, the relative stereochemistry of the hydroxyl group at C-17 remains unknown. To establish the relative stereochemistry at C-15 and C-17 in the acyclic chain, we employed *J*-based configuration analysis (*J*BCA). Long-range heteronuclear coupling constants (^3^*J*_CH_ and ^2^*J*_CH_ values) were acquired using a hetero-half-filtered TOCSY experiment [31]. Distinguishable methylene protons at C-16 allowed consecutive *J*BCA from C-15 to C-17 through C-16. Small ^3^*J*_H-15, C-17_ (1.7 Hz), small ^2^*J*_C-15, H-16a_ (0.6 Hz), and large ^2^*J*_C-15, H-16b_ (6.3 Hz) indicated that the hydroxyl group at C-15 and one of the methylene protons at C-16 in the lower field (H-16a) were in an *anti* orientation (Figure 3b), whereas small ^3^*J*_H-17, C-15_ (2.9 Hz), large ^2^*J*_C-17, H-16a_ (6.6 Hz), and small ^2^*J*_C-17, H-16b_ (0.7 Hz) suggested that the hydroxyl groups at C-17 and H-16a were in the same orientation (Figure 3c). Thus, the hydroxyl groups at C-15 and C-17 were determined as having opposite orientations. This result is consistent with the assignment predicted by the application of Kishi’s universal NMR database [32]. The relative stereochemistry of the diol groups at C-14 and C-15 established by *J*BCA was in agreement with that of the chemical reaction for its acetonide derivative. The small values of ^2^*J*_C-14, H-15_ (0.8 Hz) and ^2^*J*_C-15, H-14_ (1.8 Hz) indicated a threo configuration of both hydroxyl groups at C-14 and C-15 (Figure 3a), which agreed with that obtained from the acetonide reaction.

To determine the absolute configurations of C-14, C-15, and C-17 in **1**, we attempted a modified Mosher’s method [33]. Acetonide product **1a** was treated with *R*- and *S*-MTPA-Cl to obtain the *S*- and *R*-MTPA derivatives **1b** and **1c**, respectively. The interpretation of ^1^H NMR and ^1^H-^1^H COSY NMR of these MTPA ester derivatives allowed the assignment of the absolute configuration of C-17. A positive ∆*δ*_S-R_ value for H_3_-18 and negative ∆*δ*_S-R_ values for H-15 and H_2_-16 indicated that C-17 had an *R* configuration, subsequently establishing both *S* configurations for C-14 and C-15. The relative configurations of the consecutive stereogenic centers in compound **2** were established by *J*BCA together with the acetonide reaction. Colletogloeic acid B (**2**) was derivatized with 2,2-dimethoxy propane to obtain the acetonide derivative **2a**. HMBC data together with 1D NMR for **2a** indicated that two 1,3-acetonide groups were attached at two positions; one at between C-12 and C-14, the other at between C-15 and C-17 (Figure 4C). Both 1,3-diol groups corresponding to C-12/C-14 and C-15/C-17 were identified as a *syn* configuration based on the ^13^C resonances for methyl groups of the acetonide derivatives at *δ*_C_ 30.4/20.23 and *δ*_C_ 30.5/20.17, respectively [34]. We then established the relative configuration of diols at C-14 and C-15 by *J*BCA, and the vicinal coupling constants for H-14 and H-15 (5.5 Hz) were obtained via decoupling experiments. Small values of ^3^*J*_C-13, H-15_ (2.24 Hz), ^2^*J*_C-14, H-15_ (0.4 Hz), and ^2^*J*_C-15, H-14_ (3.2 Hz) indicated a *syn* relationship between H-14 and H-15 (Figure 3d), suggesting that all the hydroxyl groups in the side chain of **3** had *syn* configurations. The absolute stereochemistry of **3** could not be established since its *R*- and *S*-MTPA esters were decomposed. Compounds **4**, **5**, and **7** with a 1,2,4-triol system in the aliphatic chain were proposed to have the same stereochemistry as **1** based on the NMR shifts, coupling constants, and their synthetic origin.

### 2.3. Antifungal Activity

We evaluated the antifungal activities of the isolated polyketides by measuring their MIC values based on EUCAST guidelines (Figure 5). Among the isolated compounds, **8** and **9** showed potent in vitro antifungal activities against *C. neoformans* H99, with MIC values (12.5 µM) almost equivalent to those of amphotericin B (AMB) and fluconazole (FCZ) (12.5 µM), which are clinically used for the treatment of systemic cryptococcosis. By contrast, only compound **9** showed weak antifungal activities (MIC: 200 µM) against *C. albicans* SC5314, which was much less potent than AMB (12.5 µM) and FCZ (25 µM) used as positive controls. Collectively, these results indicated that compounds **8** and **9** displayed potent in vitro antifungal activity.

## 3. Materials and Methods

### 3.1. General Experimental Procedures

Optical rotation was measured at room temperature on a JASCO P-2000 polarimeter (JASCO, Easton, PA, USA) using a 1-cm cell. Infrared (IR) spectra were recorded on a Cary 630 FTIR spectrometer (Agilent Technologies, Santa Clara, CA, USA). High-resolution electrospray ionization mass spectrometry (HRESIMS) data were acquired on a UHR ESI Q-TOF mass spectrometer (Bruker, Billerica, MA, USA). Nuclear magnetic resonance (NMR) spectra were obtained using Varian NMR systems at 500 MHz (^1^H, 500 MHz; ^13^C, 125 MHz; Varian, Palo Alto, CA, USA) and a magnet system (800/45 ASCEND; ^1^H, 800 MHz; ^13^C, 200 MHz; Bruker, Billerica, MA, USA) with deuterated methanol (CD_3_OD; Cambridge Isotope Laboratories, Inc., Tewksbury, MA, USA).

### 3.2. Fungal Strain

The fungal strain JS419 was isolated from *S. japonica* Makino, which was collected from a swamp in Suncheon, South Korea, in September 2011. The tissues of this plant were cut into small pieces (0.5 × 0.5 cm) and rinsed sequentially for 1 min with 2% sodium hypochlorite, 70% ethanol, and sterilized distilled water to remove external microorganisms. After air drying, the sterilized plant tissues were incubated on malt extract agar (MEA) medium (4 g yeast extract, 10 g malt extract, 4 g potato dextrose broth, and 18 g agar per 1 L sterilized distilled water) with 50 ppm kanamycin, 50 ppm chloramphenicol, and 50 ppm Rose Bengal at 22 °C for 7 days. The actively growing fungus was transferred onto potato dextrose agar (PDA) medium (24 g potato dextrose broth and 18 g agar per 1 L sterilized distilled water). The fungal strain was identified as *C. gloeosporioides* based on the internal transcribed spacer sequences by one of the authors (S. K.) and was deposited as a 20% glycerol stock in a liquid nitrogen tank at the Wildlife Genetic Resources Bank of the National Institute of Biological Resources (Incheon, Korea).

### 3.3. Cultivation and Extraction of the Fungal Strain

The JS419 strain was cultured on a solid PDA medium for 7 days at room temperature. Agar plugs were cut into small pieces (0.5 × 0.5 cm) under aseptic conditions and inoculated on a solid rice medium (60 g rice, 90 mL distilled water per 500 mL Erlenmeyer flask). After incubation at room temperature for 30 days, 200 mL of ethyl acetate was added to each culture flask (50 × 500 mL Erlenmeyer flasks) and placed for 1 day prior to extraction. They were then filtered to separate the supernatants from the solid mycelia, after which the supernatants were evaporated under reduced pressure at 35 °C to obtain the extract (110 g).

### 3.4. Isolation of Compounds

The crude extract (110 g) was fractionated by vacuum liquid column chromatography over silica gel using a stepwise gradient of n-hexane/acetone/MeOH to obtain RM01 to RM10 fractions. The RM6 fraction (1.65 g) was further separated into RM6A to RM6J fractions using silica gel vacuum liquid column chromatography with the elution of CHCl_3_/acetone gradient solvents. Fraction RM6E (93 mg) was subjected to C18 column chromatography using a gradient of aqueous MeOH to yield compound **8** (24 mg). The RM7 fraction (2.79 g) was subjected to silica gel vacuum liquid column chromatography to obtain RM7A to RM7J fractions using a gradient of n-hexane/ethyl acetate/MeOH solvents. Fraction RM7D (302 mg) was subjected to C18 column chromatography using a gradient of aqueous MeOH, followed by purification using reversed-phase high-performance liquid chromatography (HPLC; Luna C18 (2), 5 µm, 250 × 10.0 mm; Phenomenex, Torance CA, USA; HPLC conditions: room temperature, 2.0 mL/min, 35–100% aqueous acetonitrile (ACN), and UV 210 nm) to obtain compound 10 (21.8 mg, t_R_ = 25 min). The RM8 fraction (2.02 g) was separated by silica gel column chromatography to obtain RM8A to RM8L fractions by elution with a gradient of CHCl_3_/acetone/MeOH. Fraction RM8F (539 mg) was subjected to C18 vacuum liquid column chromatography using a gradient of aqueous MeOH from 20% to 100% to obtain the RM8F1 to RM8F6 fractions. Fraction RM8F5 (30 mg) was purified by reversed-phase HPLC (Luna C18 (2), 5 µm, 250 × 10.0 mm; Phenomenex; HPLC conditions: room temperature, 2.0 mL/min, isocratic 23% aqueous ACN, and UV 210 and 254 nm) to obtain compound **3** (2.7 mg, t_R_ = 28 min). To obtain compound **5** (2.7 mg, t_R_ = 13.5 min), fraction RM8I (166 mg) was separated by C18 vacuum liquid column chromatography using a gradient of aqueous MeOH (40%, 50%, 60%, and 100%, *v*/*v*) to obtain fractions RM8I1 to RM8I5, followed by purification of the RM8I2 fraction with reversed-phase HPLC (Luna C18 (2), 5 µm, 250 × 10.0 mm; Phenomenex; HPLC conditions: room temperature, 2.0 mL/min, isocratic 30% aqueous ACN, and UV 210 nm). The RM9 fraction (2.41 g) was separated from RM9A through RM9K by silica gel vacuum liquid column chromatography using a gradient of n-hexane/ethyl acetate/MeOH. Fraction RM9F (226 mg) was further separated by C18 vacuum liquid column chromatography using a gradient of aqueous MeOH to obtain the RM9F1 to RM9F5 fractions. Fraction RM9F2 (57 mg) was further purified by reversed-phase HPLC (Luna C18 (2), 5 µm, 250 × 10.0 mm; Phenomenex; HPLC conditions: room temperature, 2.0 mL/min, 45%–70% aqueous AN, and UV 210 nm) to obtain compounds **1** (16 mg, t_R_ = 26.0 min) and **6** (7.5 mg, t_R_ = 26.0 min). To yield compound **5** (1.6 mg, t_R_ = 24.0 min), fraction RM9F3 (22 mg) was subjected to Sephadex LH-20 column chromatography eluting an isocratic 100% MeOH, followed by reverse-phase HPLC (Luna C18 (2), 5 µm, 250 × 10.0 mm; Phenomenex; HPLC conditions: room temperature, 2.0 mL/min, and 30%–50% aqueous can, and UV 210 nm). Fraction RM9G (682 mg) was separated by C18 vacuum liquid column chromatography using a stepwise gradient of aqueous MeOH (25%, 30%, 40%, 50%, 60%, and 100%, *v*/*v*) to obtain RM9G1 to RM9G6 fractions. To obtain compounds **2** (3.7 mg, t_R_ = 13.0 min) and **7** (3.2 mg, t_R_ = 23.0 min), fraction RM9G3 (42 mg) was purified by column chromatography over a Sephadex LH-20 column, followed by reverse-phase HPLC (Luna C18 (2), 5 µm, 250 × 10.0 mm; Phenomenex; HPLC conditions: room temperature, 2.0 mL/min, isocratic 30% aqueous ACN, and UV 254 nm). To obtain compound **4** (6.8 mg, t_R_ = 18 min), fraction RM9H4 (48 mg) was purified using reverse-phase HPLC (Luna C18 (2), 5 µm, 250 × 10.0 mm; Phenomenex; HPLC conditions: room temperature, 2.1 mL/min, isocratic 30% aqueous ACN, and UV 210 nm).

Colletogloeic acid A (**1**): yellowish amorphous solid; IR ν_max_ 3300, 1630, 1010 cm^−1^; UV (MeOH) λ_max_ (log ε) 244 (4.20), 320 (3.38) nm; for ^1^H and ^13^C NMR data (500 MHz in CD_3_OD), see Table 1; HMBC correlations (CD_3_OD, H-#*→*C-#) H-4→C-3 and C-5; H-6→C-4 and C-8; H-8→C-2, C-6, C-7, C-9, and C-10; H-10→C-8, C-9, C-11, and C-12; H_2_-11→C-9, C-10, C-12, and C-13; H-12a→C-11 and C-13; H-12b→C-10, C-11, C-13, and C-14; H-13a→C-11 and C-14; H-13b→C-12 and C-14; H-14→C-12, C-15, and C-16; H-15→C-13, C-14, C-16, and C-17; H-16a→C-14, C-15, C-17, and C-18; H-16b→C-15, C-17, and C-18; H-17→C-15 and C-16; H_3_-18→C-16 and C-17; (+)HRESIMS *m/z* 375.1420 [M + Na − H_2_O]^+^ (calcd for C_18_H_24_O_7_Na, 375.1414).

Colletogloeic acid B (**2**): yellowish amorphous solid; [α]_D_^25^ = −9.9 (*c* 0.1, MeOH); IR ν_max_ 3300, 1640, 1020 cm^−1^; UV (MeOH) λ_max_ 244, 326 nm; for ^1^H and ^13^C NMR data (500 MHz in CD_3_OD), see Table 1; HMBCs (CD_3_OD, H-#→C-#) H-4→C-1, C-2, C-3, C-5, and C-6; H-8→C-2, C-6, C-9, and C-10; H-10a and H-10b→C-8, C-9, C-11, and C-12; H-11a→C-10; H-11b→C-12; H-13a→C-11, C-12, and C-15; H-16a→C-14, C-17, and C-18; H-16b→C-17 and C-18; H-17→C-15 and C-16; H_3_-18→C-16 and C-17; (+)HRESIMS *m/z* 425.1010 [M + Na − H_2_O]^+^ (calcd for C_18_H_23_O_8_ClNa, 425.0974).

Colletogloeic acid C (**3**): yellowish amorphous solid; IR ν_max_ 3320, 2940, 2820, 1020 cm^−1^; UV (MeOH) λ_max_ (log ε) 244 (4.00), 328 (3.21) nm; for ^1^H and ^13^C NMR data (500 MHz in CD_3_OD), see Table 1; HMBCs (CD_3_OD, H-#→C-#) H-4→C-2, C-5, and C-6; H-6→C-2, C-4, and C-8; H-8→C-2, C-6, C-9, and C-10; H-10→C-8, C-9, C-11, and C-12; H-11→C-9, C-12, and C-13; H-12→C-10, C-11, C-13, and C-14; H-16a→C-14, C-15, and C-17; H_3_-18→C-16 and C-17; (+)HRESIMS *m/z* 357.1312 [M + Na − H_2_O]^+^ (calcd for C_18_H_22_O_6_Na, 357.1309).

Colletoresorcylic lactone (**4**): yellowish amorphous solid; [α]_D_^25^ = −5.1 (*c* 0.1, MeOH); IR ν_max_ 3380, 1680, 1620 cm^−1^; UV (MeOH) λ_max_ 211, 262, 298 nm; for ^1^H and ^13^C NMR data (500 MHz in CD_3_OD), see Table 2; HMBCs (CD_3_OD, H-#→C-#) H-4→C-2, C-3, C-5, and C-6; H-6→C-2, C-4, C-5, and C-8; H-8a and H-8b→C-2, C-6, C-7, and C-9; H-10a and H-10b→C-9, C-11, and C-12; H-12→C-13; H-13a→C-11, C-12, C-14, and C-16; H-13b→C-11, C-12, and C-14; H-14→C-12, C-13, C-15, and C-16; H-15→C-1, C-14, and C-17; H-16a→C-14, C-17, and C-18; H-16b→C-14, C-15, and C-18; H_3_-18→C-16 and C-17; (+)HRESIMS *m/z* 375.1419 [M + Na]^+^ (calcd for C_18_H_24_O_7_Na, 375.1414).

Nordinonediol chloride (**5**): yellowish amorphous solid; [α]_D_^25^ = −21.7 (*c* 0.1, MeOH); IR ν_max_ 3300, 1640, 1240 cm^−1^; UV (MeOH) λ_max_ (log ε) 220 (3.85), 260 (3.74), 310 (3.34) nm; for ^1^H and ^13^C NMR data (500 MHz in CD_3_OD), see Table 2; HMBCs (CD_3_OD, H-#→C-#) H-4→C-1, C-2, C-3, and C-5; H-8a and H-8b→C-2, C-6, and C-9; H-10a→C-11; H-10b→C-9; H-14→C-15 and C-16; H-15→C-14, C-16, and C-17; H-16a→C-17 and C-18; H-16b→C-15 and C-18; H-17→C-1, C-16, and C-18; H_3_-18→C-16 and C-17; (+)HRESIMS *m/z* 409.1041 [M + Na]^+^ (calcd for C_18_H_23_ClO_7_Na, 409.1025).

Hydroxymonocillin I glycoside (**7**): yellowish amorphous solid; [α]_D_^25^ = +4.5 (*c* 0.1, MeOH); IR ν_max_ 3300, 1650, 1010 cm^−1^; UV (MeOH) λ_max_ (log ε) 220 (3.70), 258 (3.52), 308 (3.25) nm; for ^1^H and ^13^C NMR data (500 MHz in CD_3_OD), see Table 2; HMBCs (CD_3_OD, H-#→C-#) H-4→C-2, C-3, C-5, and C-6; H-8a and H-8b→C-2, C-7, and C-9; H-10→C-12; H-12→C-10; H-17→C-1, C-16, and C-15; H_3_-18→C-16 and C-17; H-1′→C-5, C-2′, C-3′, and C-4′; (+)HRESIMS *m/z* 519.1033 [M + Na]^+^ (calcd for C_23_H_28_O_12_Na, 519.1036).

### 3.5. Preparation of Acetonide Derivatives of **1** (**1a**) and **2** (**2a**)

The thoroughly dried compound **1** (4.5 mg) was dissolved in anhydrous solution (MeOH:CH_2_Cl_2_ = 1:1, 9 mL) and treated with 2,2-dimethoxypropane (4.5 mL) and pyridinium *p*-toluenesulfonate (9 mg) in a reaction vial. After reaction on a stirrer at room temperature for 6 h, saturated NaHCO_3_ (20 μL) was added to quench the reaction. The mixture was purified by reversed-phase HPLC (Luna C18 (2), 5 µm, 250 × 10.0 mm; Phenomenex; HPLC conditions: room temperature, 2.0 mL/min, and UV 210 and 254 nm) using gradient elution from 35% to 55% in ACN/H_2_O for 30 min. The final product (**1a**, 2.3 mg) was eluted at a retention time of 29 min under HPLC conditions.

Colletogloeic acid A acetonide derivative (**1a**): ^1^H NMR (500 MHz, CD_3_OD) *δ*_H_ 6.34 (1H, s, H-8), 6.30 (2H, d, *J* = 2.5 Hz, H-4 and H-6), 4.30 (1H, ddd, *J* = 10.5, 6.0, 3.0 Hz, H-15), 4.09 (1H, ddd, *J* = 9.0, 5.5, 4.0 Hz, H-14), 3.92 (1H, m, H-17), 2.54 (1H, t, *J* = 6.5 Hz, H-10), 1.74 (2H, m, H-11), 1.61 (1H, m, H-12a), 1.55 (1H, m, H-16a), 1.50 (2H, m, H_2_-13), 1.46 (1H, m, H-16b), 1.42 (1H, m, H-12b), 1.38 (3H, s, H_3_-20), 1.31 (3H, s, H_3_-21), 1.19 (3H, d, *J* = 6.5 Hz, H_3_-18); ^13^C NMR (125 MHz, CD_3_OD) *δ*_C_ 168.1 (C-1), 167.6 (C-3), 165.0 (C-5), 158.9 (C-9), 141.5 (C-7), 108.8 (C-19), 105.4 (C-8), 103.8 (C-4), 102.7 (C-6), 99.8 (C-2), 79.2 (C-14), 76.2 (C-15), 65.7 (C-17), 40.2 (C-16), 34.1 (C-10), 30.8 (C-13), 29.1 (C-20), 28.0 (C-11), 26.8 (C-12), 26.3 (C-21), 24.8 (C-18); HMBCs (CD_3_OD, H-#→C-#) H-4→C-2, C-3, C-5, and C-6; H-6→C-4, C-5, and C-8; H-8→C-2, C-7, C-9, and C-10; H-10→C-8, C-9, C-11, and C-12; H-11→C-9, C-10, C-12, and C-13; H-12a→C-13; H-13→C-12 and C-14; H-14→C-12, C-16, and C-20; H-15→C-13, C-17, and C-20; H-16a→C-14, C-15, C-17, and C-18; H-16b→C-17 and C-18; H-17→C-15 and C-16; H_3_-18→C-16 and C-17; H_3_-20→C-19 and C-21; H_3_-21→C-19 and C-20; ESIMS *m/z* 415.4 [M + Na − H_2_O]^+^

Colletogloeic acid B diacetonide derivative (**2a**): ^1^H NMR (500 MHz, CD_3_OD) *δ*_H_ 6.70 (1H, s, H-8), 6.42 (1H, s, H-4), 4.02 (1H, m, H-17), 3.94 (1H, m, H-12), 3.73 (2H, m, H-14 and H-15), 2.67 (2H, m, H_2_-10), 1.85 (1H, m, H-11a), 1.80 (1H, m, H-11b), 1.71 (1H, m, H-13a), 1.68 (1H, m, H-16a), 1.43, 1.41, 1.31, and 1.30 (each 3H, s, 4 × acetonide methyl), 1.19 (3H, d, *J* = 6.0 Hz, H_3_-18), 1.18 (1H, m, H-13b), 1.08 (1H, m, H-16b); ^13^C NMR (125 MHz, CD_3_OD) *δ*_C_ 167.5 (C-1), 165.4 (C-3), 163.3 (C-5), 160.1 (C-9), 137.9 (C-7), 107.4 (C-6), 103.2 (C-4), 102.1 (C-8), 100.1 (two acetonide ketal carbons), 100.0 (C-2), 73.5 (C-15), 73.4 (C-14), 69.2 (C-12), 66.6 (C-17), 36.2 (C-16), 34.7 (C-11), 34.5 (C-13), 30.6 (C-10), 30.5, 30.4, 20.23, and 20.17 (four acetonide methyl), 22.7 (C-18); HMBCs (CD_3_OD, H-#→C-#) H-4→C-2, C-3, and C-6; H-8→C-2, C-9, and C-10; H-10→C-8, C-9, C-11, and C-12; H-11a and H-11b→C-9, C-10, C-12, and C-13; H-14→C-15 and C-16; H-15→C-14 and C-16; H_3_-18→C-16 and C-17; acetonide methyl protons (*δ*_H_ 1.43/1.41)→acetonide ketal and acetonide methyl carbon (*δ*_C_ 30.4/30.5); acetonide methyl protons (*δ*_H_ 1.31/1.30)→acetonide ketal and acetonide methyl carbon (*δ*_C_ 20.23/20.17); H_3_-21→C-15, C-17, C-19, and C-20; H_3_-23→C-22 and C-24; H_3_-24→C-12, C-14, C-22, and C-23; ESIMS *m/z* 505.4 [M + Na − H_2_O]^+^.

### 3.6. Preparation of (R)-and (S)-α-Methoxy-α-trifluoromethylphenylacetic Acid (MTPA) Ester Derivatives of **1** (**1b**/**1c**)

The acetonide product of **1** (**1a**) was prepared in two vials and evaporated to dryness under a high vacuum for 24 h. Each sample was dissolved in anhydrous pyridine (600 μL), sealed completely, and allowed to stand at room temperature for 5 min, followed by the addition of 30 μL of (*R*)- and (*S*)-MTPA chloride (MTPA-Cl) to the reaction vials. After reaction for 18.5 h under argon, to quench the reaction, anhydrous MeOH (50 μL) was added. The mixture was purified by reversed-phase HPLC (Luna C18 (2), 5 µm, 250 × 10.0 mm; Phenomenex; HPLC conditions: room temperature, 2.0 mL/min, and UV 210 and 254 nm) using gradient elution (from 10% to 100%) in ACN/H_2_O for 70 min. Under the HPLC conditions described above, the *S*-MTPA (**1b**) and *R*-MTPA (**1c**) esters eluted at retention times of 62 min and 61 min, respectively. The molecular formula was confirmed as C_51_H_51_F_9_O_14_ using (+) ESI-MS ([M + Na − H_2_O]^+^
*m/z* at 1063.6).

*S*-MTPA ester (**1b**): ^1^H NMR (500 MHz, CD_3_OD) *δ*_H_ 7.75-7.39 (15H, m, MTPA-Ar), 7.38 (1H, m, H-4), 6.83 (1H, d, *J* = 2.0 Hz, H-6), 6.59 (1H, s, H-8), 5.28 (1H, m, H-17), 3.86 (1H, m, H-14), 3.77 (3H, s, MTPA-OCH_3_), 3.75 (1H, m, H-15), 3.70 (3H, s, MTPA-OCH_3_), 3.54 (3H, s, MTPA-OCH_3_), 2.57 (2H, t, *J* = 7.0 Hz, H-10), 1.73 (2H, m, H-11), 1.65 (2H, m, H-16), 1.51 (1H, m, H-12a), 1.46 (1H, m, H-13a), 1.33 (1H, m, H-13b), 1.29 (1H, m, H-12b), 1.38 (3H, d, *J* = 6.5 Hz, H-18), 1.37 (3H, s, H-20), 1.24 (3H, s, H-21).

*R*-MTPA ester (**1c**): ^1^H NMR (500 MHz, CD_3_OD) *δ*_H_ 7.75-7.40 (15H, m, MTPA-Ar), 7.39 (1H, d, *J* = 2.0 Hz, H-4), 6.83 (1H, d, *J* = 2.0 Hz, H-6), 6.60 (1H, s, H-8), 5.26 (1H, m, H-17), 4.10 (1H, m, H-14, 15), 3.77 (3H, s, MTPA-OCH_3_), 3.70 (3H, s, MTPA-OCH_3_), 3.50 (3H, s, MTPA-OCH_3_), 2.59 (2H, t, J = 7.5 Hz, H-10), 1.76 (2H, m, H-11), 1.69 (2H, m, H-16), 1.40 (3H, s, H-20), 1.31 (3H, d, J = 6.5 Hz, H-18), 1.29 (3H, s, H-21).

### 3.7. Antifungal Activity Assay

We determined the minimum inhibitory concentration (MIC) of the selected compounds according to European Committee on Antimicrobial Susceptibility Testing (EUCAST) guidelines using the E.Def 7.1 methodology. *C. albicans* (SC5314) and *C. neoformans* (H99) strains were cultured overnight at 30 °C in YPD liquid medium, washed twice with sterilized H_2_O, and then resuspended in sterilized H_2_O. We then added 100 µL of the cell suspension, in which the cell concentration was adjusted to an OD_600_ of 1.0, to 10 mL of 3-(*N*-morpholino)propanesulfonic acid (MOPS)-buffered Roswell Park Memorial Institute (RPMI)-1640 medium (pH 7.4; with 0.165 M MOPS and 2% glucose for EUCAST) and loaded onto 96-well plates containing two-fold serially-diluted compounds. The final concentration of the tested compounds ranged from 1.56 µM to 200 µM. The 96-well plates were incubated at 35 °C for 2 days, and the cell density in each well was measured at OD595 [35].

## 4. Conclusions

In this study, we isolated nine β-resorcylic acid derivatives, represented by RALs, from cultures of *C. gloeosporioides* JS0419. While most RALs reported to date feature a 14-membered macrocyclic ring fused to β-resorcylic acid, compounds **1**–**3** harbored β-resorcylic acid with a long unesterified aliphatic side chain and compound **4** had a 12-membered macrocyclic ring fused to the β-resorcylic acid in this chemical investigation. Although several ring-opened β-resorcylic acids have been reported to date [36,37], they mostly had non-oxidized C-9 in the long aliphatic chain. Given the rarity of this observation in polyketides, we did not exclude the possibility of artifacts. For subsequent analyses, we did not use acids or alkalis throughout the isolation and purification procedures. We carefully analyzed the HPLC chromatograms of the initial extracts of this fungal strain, which identified a fair amount of resorcylic acid with a free chain together with RAL-type compounds. Compound **4** was lactonized with the ester linkage between C-15 and C-1 to form a 12-membered lactone ring. Although many RALs had 12-membered rings, to our knowledge, this is the first report of RAL with not methyl but propyl group at C-15. Although many RAL-type compounds have been evaluated for antifungal activities against several plant pathogenic fungi, to our knowledge, there is no report on their activity against *C. albicans* and *C. neoformans* causing fatal candidiasis and cryptococcosis, respectively. The known compounds, radicicol, and monocillin I, exhibit potent in vitro antifungal activities, suggesting that an epoxide group, non-sugar moiety, and lactonization of the long polyketide chain could be important for antifungal activities. Optimization of the polyketides through structure–activity relationship studies may further increase their antifungal activity and allow the examination of their in vivo efficacy for treating systemic fungal infections.

## Figures and Tables

**Figure 1 marinedrugs-20-00195-f001:**
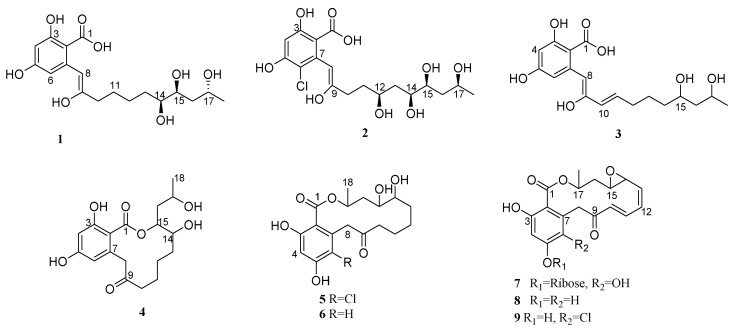
The structure of isolated compounds **1**–**9**.

**Figure 2 marinedrugs-20-00195-f002:**
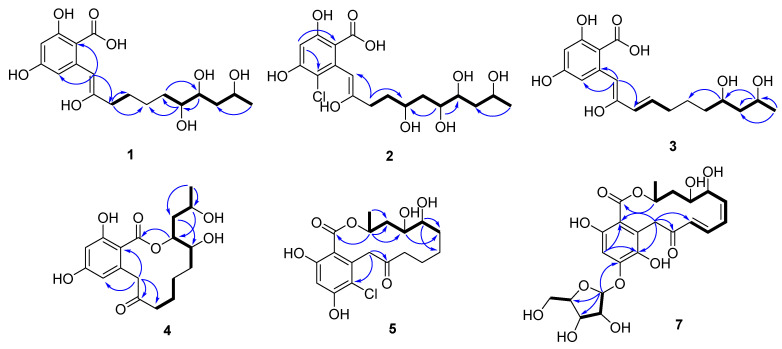
Key ^1^H-^1^H COSY (bold) and HMBCs (arrow) for compounds **1**–**5**, and **7**.

**Figure 3 marinedrugs-20-00195-f003:**
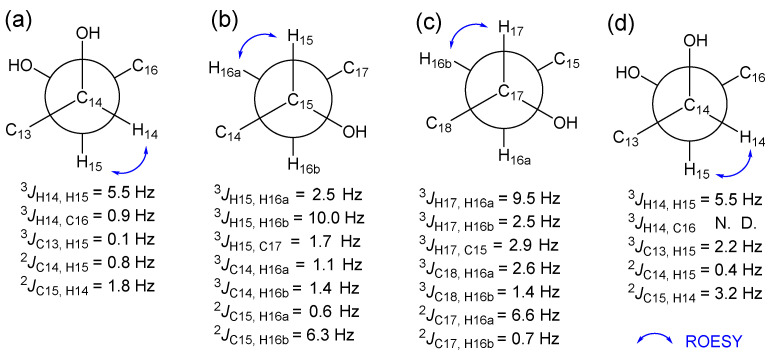
*J*-based configuration analysis. (**a**) C-14 and C-15, (**b**) C-15 and C-16, (**c**) C-16 and C-17 in compound **1,** and (**d**) C-14 and C-15 in compound **2**.

**Figure 4 marinedrugs-20-00195-f004:**
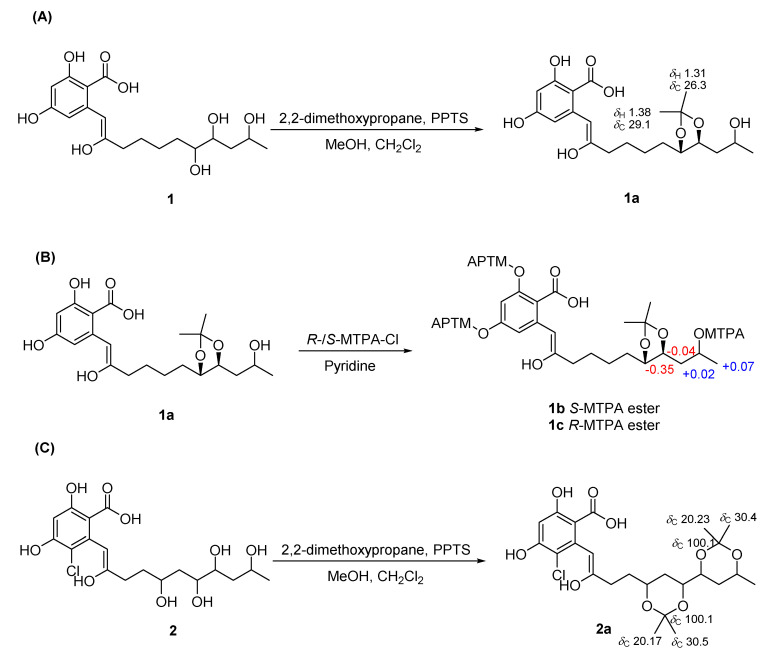
Scheme for chemical derivatization to determine relative and absolute configuration of compounds **1** and **2**. (**A**) Synthesis of 1,2−diol−acetonide for **1** and its ^1^H and ^13^C NMR chemical shifts in CD_3_OD (**1a**). (**B**) Synthesis of *S*− and *R*−MTPA esters (**1b** and **1c**) for the acetonide derivative (**1a**) and its ∆*δ*_S__−R_ values observed in ^1^H NMR (CD_3_OD). (**C**) Synthesis of 1,2−diol−acetonide for **2** and its ^13^C shifts in CD_3_OD (**2a**).

**Figure 5 marinedrugs-20-00195-f005:**
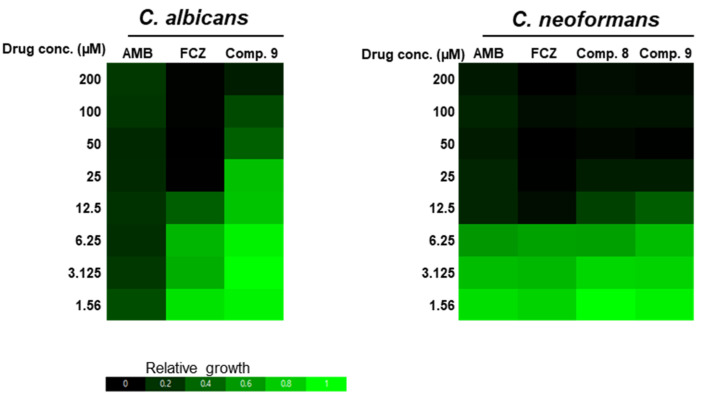
Antifungal activities of compounds **8** and **9**. The heat map of EUCAST MIC test results for compounds **8** and **9** against *C. albicans* SC5314 and *C. neoformans* H99. Fungal cells were prepared, as described, and incubated at 35 °C for 2 days in 96-well microtiter plates containing MOPS-buffered RPMI-1640 medium with two-fold-diluted natural compounds. Amphotericin B (AMB) and fluconazole (FCZ) were used as positive controls.

**Table 1 marinedrugs-20-00195-t001:** ^1^H and ^13^C NMR data for compounds **1**–**3** in CD_3_OD.

No.	1		2		3	
	*δ*_C,_ Type ^b^	*δ*_H_ (*J* in Hz) ^a^	*δ*_C,_ Type ^b^	*δ*_H_ (*J* in Hz) ^a^	*δ*_C,_ Type ^d^	*δ*_H_ (*J* in Hz) ^c^
1	168.1, C		167.6, C		167.6, C	
2	99.6, C		99.0, C		100.0, C	
3	164.8, C		165.6, C		167.3, C	
4	102.8, CH	6.29, d (2.0)	103.8, CH	6.42, s	103.0, CH	6.31, d (2.0)
5	167.7, C		163.3, C		164.9, C	
6	103.9, CH	6.28, d (2.0)	108.3, C		104.6, CH	6.34, d (2.0)
7	141.5, C		137.5, C		141.5, C	
8	105.4, CH	6.31, s	101.9, CH	6.72, s	106.0, CH	6.37, s
9	159.0, C		159.9, C		153.5, C	
10	34.1, CH_2_	2.52, t (7.5)	30.8, CH_2_	2.75, ddd (15.0, 9.5, 5.0) 2.67, dd (9.5, 7.0)	123.4, CH	6.16, d (15.5)
11	28.1, CH_2_	1.71, m	35.5, CH_2_	1.96, m 1.64, m	137.4, CH	6.54, dt (15.5, 2.5)
12	26.5, CH_2_	1.63, m 1.44, m	70.4, CH	3.91, dq (7.0, 3.5)	33.8, CH_2_	2.28, q (7.0)
13	33.4, CH_2_	1.65, m 1.42, m	40.3, CH_2_	1.81, dt (8.5, 3.0) 1.64, m	26.1, CH_2_	1.65, m 1.54, m
14	76.1, CH	3.42, ddd (8.0, 5.5, 2.5)	74.8, CH	3.61, m	38.4, CH_2_	1.51, m
15	72.9, CH	3.66, ddd (9.8, 5.5, 2.5)	75.0, CH	3.61, m	71.3, CH	3.75, m
16	42.4, CH_2_	1.61, m 1.48, ddd (14, 9.8, 2.8)	42.0, CH_2_	1.69, ddd (14.5, 5.5, 3.0) 1.60, td (7.0, 2.5)	46.9, CH_2_	1.60, m 1.56, m
17	65.6, CH	4.00, dqd (9.5, 6.0, 2.8)	65.6, CH	4.00, dq (12.5, 6.0)	67.7, CH	3.95, m
18	24.6, CH_3_	1.20, d (6.0)	23.6, CH_3_	1.19, d (6.0)	23.8, CH_3_	1.18, d (6.0)

^a^ Measured at 500 MHz. ^b^ Measured at 125 MHz. ^c^ Measured at 800 MHz. ^d^ Measured at 200 MHz.

**Table 2 marinedrugs-20-00195-t002:** ^1^H and ^13^C NMR data for compounds **4**, **5**, and **7** in CD_3_OD.

No.	4		5		7	
	*δ*_C,_ Type ^b^	*δ*_H_ (*J* in Hz) ^a^	*δ*_C,_ Type ^b^	*δ*_H_ (*J* in Hz) ^a^	*δ*_C,_ Type ^b^	*δ*_H_ (*J* in Hz) ^a^
1	171.9, C		172.5, C		174.8, C	
2	107.5, C		107.1, C		117.1, C	
3	165.5, C		163.5, C		168.5, C	
4	103.0, CH	6.26 (d, 2.5)	104.2, C		105.3, CH	6.87 (s)
5	163.9, C		160.9, C		156.7, C	
6	113.9, CH	6.14 (d, 2.5)	117.2, CH	6.41 (s)	157.3, C	
7	140.3, C		137.6, C		134.3, C	
8	51.4, CH_2_	4.48 (d, 18.5) 3.82 (d, 18.5)	48.3, CH_2_	4.84 (d, 18.5) 4.25 (d, 18.5)	46.4, CH_2_	4.01 (d, 16.0) 3.93 (d, 16.0)
9	212.5, C		208.7, C		199.2, C	
10	42.5, CH_2_	2.70 (ddd, 16.0, 9.5, 2.0) 2.38 (ddd, 16.0, 9.5, 2.3)	39.2, CH_2_	2.98 (ddd, 18.5, 12.0, 3.5) 2.55 (dt, 19.0, 3.5)	131.8, CH	6.08 (d, 16.0)
11	23.1, CH_2_	1.98 (m) 1.82 (m)	22.6, CH_2_	1.93 (m) 1.33 (m)	140.8, CH	7.59 (dd, 16.0, 10.0)
12	25.5, CH_2_	1.58 (m)	23.1, CH_2_	1.53 (m) 1.21 (dd, 11.5, 6.0)	131.1, CH	6.22 (t, 10.0)
13	32.4, CH_2_	1.54 (m) 1.45 (m)	31.4, CH_2_	1.53 (m) 1.37 (m)	137.4, CH	5.77 (dd, 10.0, 4.5)
14	73.1, CH	3.73 (ddd, 9.0, 7.0, 2.5)	76.4, CH	3.59 (dt, 11.0, 2.5)	56.6, CH	3.36 (m)
15	77.6, CH	4.97 (ddd, 9.0, 7.5, 4.0)	69.3, CH	3.49 (br d, 10.5)	56.9, CH	3.08 (dt, 8.5, 3.0)
16	42.4, CH_2_	2.01 (m) 1.77 (ddd, 14.5, 7.0, 3.0)	36.6, CH_2_	1.98 (dd, 15.5, 11.0) 1.72 (dd, 15.5, 11.0)	38.1, CH_2_	2.44 (dt, 14.5, 3.0) 1.66 (ddd, 14.0, 9.0, 4.5)
17	65.4, CH	3.90 (dtd, 9.5, 6.5, 3.0)	71.7, CH	5.43 (m)	72.4, CH	5.38 (m)
18	24.3, CH_3_	1.18 (d, 6.5)	21.7, CH_3_	1.41 (d, 6.0)	18.6, CH_3_	1.53 (d, 6.5)
1′					103.4, CH	5.65 (d, 4.5)
2′					73.7, CH	4.22 (dd, 6.5, 4.5)
3′					71.1, CH	4.09 (dd, 6.5, 2.5)
4′					88.3, CH	4.18 (dd, 6.5, 3.0)
5′					63.2, CH_2_	3.69 (dd, 12.5, 3.5) 3.65 (dd, 12.0, 3.0)

^a^ Measured at 500 MHz. ^b^ Measured at 200 MHz.

## Data Availability

Not applicable.

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
