# Peer review of "Rare β-Resorcylic Acid Derivatives from a Halophyte-Associated Fungus Colletotrichum gloeosporioides JS0419 and Their Antifungal Activities"

_marinedrugs, 2022, doi:10.3390/md20030195_

Round 1
Reviewer 1 Report
The manuscript submitted for review reports on the isolation of seven new β-resorcylic acid derivatives from a halophyte-associated fungus, Colletotrichum gloeosporioides JS0419. The work was carried out using all relevant methods for determining the structure of natural compounds of this class, including the determination of the configurations of asymmetric centers. The correctness of the establishment of the structure is beyond doubt.
However, there are notes:
- Figure 1: You must specify the numbering of atoms in all structures. The complete absence of numbering (structures 3-5 and 8-10), as well as insufficiently indicated numbering in the aliphatic chain of compounds 1 and 2, complicate the perception of the text.
2 Table 1: must be checked! There are errors for compounds 2 (CH2-8 and C-9) and 3 (C-6 not CH-6).
- Lines 82-83: cannot be included in the chain established by COSY CH-8 and C(O)-9, since there are no CH-8/C(O)-9/CH2-10 correlations in the spectra! (they cannot be). The chain established by COSY should start with CH2-10 and add the HMBC correlations from H-8 to C-10 and from H2-10 (or H2-11) to C-9.
- The description of the establishment of structures 3, 4, 6 and 8 must begin with a new paragraph.
- Figure 1: The structural formula of compound 3 is incorrect. In the text of article (paragraph 2.2) it is reported that all hydroxyl groups of the aliphatic chain in 3 are syn. The figure is shown differently.
- Apparently, there is also a mistake in the abstract, the authors report the establishment of seven new metabolites, not six.
- Section 2.2 seems incomplete to me: nothing is said about determining the configurations of the asymmetric centers of compounds 4, 5, 6 and 8. Why did the authors not do this? You need to write at least a couple of words.
- Experimental part, paragraph 3.5: you need to indicate the mass of compound 1 taken for the experiment and the yield of acetonide. For what second compound was the acetonide derivative obtained? The text of the article says that diacetonide was obtained for compound 3, and in the experiment for compound 4. Which is correct? It is necessary to provide correct NMR data, as well as the mass spectrum of the resulting diacetonide 3 (or acetonide 4).
Summing up, I can say that the manuscript leaves a pleasant impression, despite several shortcomings and errors. Unfortunately, the new compounds did not show antifungal activity, and the authors did not make other biological activity studies. However, the structural part of the work was done at a high level.
I think that the manuscript can be accepted for publication after correcting minor comments.
Author Response
Dear Editor,
Thank you very much for your careful review on our manuscript. According to the editor’s and reviewers’ comments, all comments were answered one by one as shown below. Thus, several parts of the revised manuscript have been corrected and indicated in red color concerning some modification in comparison with the previous manuscript.
In the revised manuscript, compound 2 in the original manuscript was removed because the structure of compound 2 was revised to compound 5 of the original manuscript) when writing the manuscript. Compound 2 should have been removed in original submission but it was not deleted by mistake. We truly apologize for causing confusion. Therefore, the numbers of the isolated compounds were revised accordingly.
Reviewer(s)' Comments to Author:
Reviewer #1:
The manuscript submitted for review reports on the isolation of seven new β-resorcylic acid derivatives from a halophyte-associated fungus, Colletotrichum gloeosporioides JS0419. The work was carried out using all relevant methods for determining the structure of natural compounds of this class, including the determination of the configurations of asymmetric centers. The correctness of the establishment of the structure is beyond doubt.
However, there are notes:
- Figure 1: You must specify the numbering of atoms in all structures. The complete absence of numbering (structures 3-5 and 8-10), as well as insufficiently indicated numbering in the aliphatic chain of compounds 1 and 2, complicate the perception of the text.
Response: The numbers of the atoms in all compounds were specified in Figure 1 as suggested by the reviewer.
- Table 1: must be checked! There are errors for compounds 2 (CH2-8 and C-9) and 3 (C-6 not CH-6).
Response: Table 1 was carefully checked and revised according to the reviewer’s comments.
- Lines 82-83: cannot be included in the chain established by COSY CH-8 and C(O)-9, since there are no CH-8/C(O)-9/CH2-10 correlations in the spectra! (they cannot be). The chain established by COSY should start with CH2-10 and add the HMBC correlations from H-8 to C-10 and from H2-10 (or H2-11) to C-9.
Response: According to the reviewer’s comments, the COSY started with CH2-10 to CH3-18 in lines 82-83.
- The description of the establishment of structures 3, 4, 6 and 8 must begin with a new paragraph.
Response: According to the reviewer’s suggestion, the description of the structures 3, 4, 5, and 7 started with a new paragraph in lines 126, 137, 148, and 152.
- Figure 1: The structural formula of compound 3 is incorrect. In the text of article (paragraph 2.2) it is reported that all hydroxyl groups of the aliphatic chain in 3 are syn. The figure is shown differently.
Response: The structure of compound 3 (compound 2 in the revised manuscript) was revised based on the results from acetonide reaction and JBCA analysis (Figure 1).
- Apparently, there is also a mistake in the abstract, the authors report the establishment of seven new metabolites, not six.
Response: We truly apologize for causing confusion. The number of new compounds is six since we did not remove the structure 2 by mistake in the original manuscript.
- Section 2.2 seems incomplete to me: nothing is said about determining the configurations of the asymmetric centers of compounds 4, 5, 6 and 8. Why did the authors not do this? You need to write at least a couple of words.
Response: The stereochemistry of the compounds 3, 4, 5, and 7 could not be established by chemical reactions due to the limited amounts of the sample. However, compounds 4, 5, and 7 with 1,2,4-triol system in the aliphatic chain were proposed to have the same stereochemistry as 1 based on the NMR shifts, coupling constants, and their synthetic origin. It was mentioned in the manuscript lines 238-240.
- Experimental part, paragraph 3.5: you need to indicate the mass of compound 1 taken for the experiment and the yield of acetonide. For what second compound was the acetonide derivative obtained? The text of the article says that diacetonide was obtained for compound 3, and in the experiment for compound 4. Which is correct? It is necessary to provide correct NMR data, as well as the mass spectrum of the resulting diacetonide 3 (or acetonide 4).
Response: As the reviewer commented, the mass of compound 1 and the yield of the acetonide were added in the manuscript (lines 395 and 401). And the diacetonide product was obtained for compound 2 in the revised manuscript. All the numbers of the isolated compounds were re-arranged since one compound was removed in the revised manuscript.
Summing up, I can say that the manuscript leaves a pleasant impression, despite several shortcomings and errors. Unfortunately, the new compounds did not show antifungal activity, and the authors did not make other biological activity studies. However, the structural part of the work was done at a high level.
I think that the manuscript can be accepted for publication after correcting minor comments.
Please contact me with any questions or concerns in regards to this submission by email at our conveniences.
Yours sincerely,
Sang Hee Shim, Ph.D./ Professor
Natural Products Research Institute, College of Pharmacy, Seoul National University
1 Gwanak-ro, Gwanak-gu, 08826 Seoul, South Korea
Tel: 82-2-880-2479
E-mail: sanghee_shim@snu.ac.kr

Reviewer 2 Report
Comments: Major revisions are required
Bang et al describes the discovery of 10 β-resorcylic acid lactones (RALs) derivatives from a halophyte-associated fungus, Colletotrichum gloeosporioides JS0419, including 7 new compounds. The planar structure, relative and absolute configurations of 1 were elucidated by NMR, chemical reaction, J-based configuration analysis (JBCA) and modified Mosher’s method. However, the absolute configurations of 3, and the stereochemistry of 4-8 were not analyzed. Besides, the elucidation of structures of new compounds were not well arranged. So, there have some concerns that should be addressed before consider to accept.
Major concerns:
- Try to assign the stereochemistry of new compounds according biosynthetic pathway. For example, 4 was a biosynthetic precursor of 1/2, the stereochemistry of 4 can be tentatively assigned to be the same as 1/2. Similarly, the stereochemistry of 3, 5, and 6 can be assigned.
- The structure elucidation part need to be rearranged.
(1) All of the HMBC and COSY figures for new compounds should be presented.
(2) It’s better to write the structure elucidation for a new compound in a new paragraph.
(3) The assignment of the long aliphatic chain is difficult to understand by interpretation of the 1H-1H COSY, because there are overlapping signals (CH2) in 1H NMR spectra. Therefore, the assignment of the long aliphatic chain needs additional supporting evidence.
(4) Page 6, line 196, “The 13C resonances for the two methyl groups of the acetonide derivateve, …indicated a syn configuration of the dihydroxyl groups at C-14 and C-15 [30]”. The method and the reference for assigning the configuration of acetonide derivative are incorrect, please refer to literatures 10.1021/jo00103a012, 10.1074/jbc.RA120.015563. Similarly, for page 7, line 232 “The 13C resonances for four methyl groups of the …” are incorrect. These configurations should be reassigned.
Minor concerns
- Page 1 line 17, “Six new β-resorcylic acid derivatives (1–6 and 8)” should be “seven new β-resorcylic acid derivatives (1–6 and 8)”
- 2. Table 1, Table 2 “δH, (J in Hz)”, should be “δH, (J in Hz)”, the data need to be rearranged accordingly.
- “[M+Na-H2O]+” should be “[M + Na - H2O]+”, please check throughout the manuscript.
- “syn configuration” should be “syn configuration”, please check all manuscript.
- “JBCA” should be “JBCA”, please check all manuscript.
- OD values for 1 and 4 are too small. These two compounds were almost optically inactive. No need to list the OD values.
- Page 15 line 421 showed the data of colletogloeic acid D diacetonide derivative (4a), which is not mentioned in the result.
- 8. Page 4, line 127, please revise "1, 3, 4-triol" as "1, 2, 4-triol".
- 9. Page 7, line 238, please check the configuration at C-17 of compound 3 shown in Figure 1. It does not have a syn relationship with C-15.
- Page 4, lines 141-143, Although compound 5 has the same quasi molecular ion peak as 1 and 2, the molecular formula of compound 5 should not be the same as those of compounds 1 and 2 according to their structures shown in Figure 1. Please make a careful check of this point.
Other concerns:
Are compounds 1 and 2 a pair of enol tautomers? Are compounds 1 and 2 stable? and how about compound 3 and 4?
Author Response
Answers of Editor’s and Reviewers’ Comments
Manuscript ID: marinedrugs-1615334
Title: Rare β-resorcylic acid derivatives from a halophyte-associated fungus
Colletotrichum gloeosporioides JS0419 and their anti-fungal activities
Authors: Sunghee Bang, Jaekyeong Kim, Jiwon Oh, Ji-Seok Kim, Seong-Ryong Yu, Stephen T. Deyrup, Yong-Sun Bahn, and Sang Hee Shim
Dear Editor,
Thank you very much for your careful review on our manuscript. According to the editor’s and reviewers’ comments, all comments were answered one by one as shown below. Thus, several parts of the revised manuscript have been corrected and indicated in red color concerning some modification in comparison with the previous manuscript.
In the revised manuscript, compound 2 in the original manuscript was removed because the structure of compound 2 was revised to compound 5 of the original manuscript) when writing the manuscript. Compound 2 should have been removed in original submission but it was not deleted by mistake. We truly apologize for causing confusion. Therefore, the numbers of the isolated compounds were revised accordingly.
Reviewer #2:
Bang et al describes the discovery of 10 β-resorcylic acid lactones (RALs) derivatives from a halophyte-associated fungus, Colletotrichum gloeosporioides JS0419, including 7 new compounds. The planar structure, relative and absolute configurations of 1 were elucidated by NMR, chemical reaction, J-based configuration analysis (JBCA) and modified Mosher’s method. However, the absolute configurations of 3, and the stereochemistry of 4-8 were not analyzed. Besides, the elucidation of structures of new compounds were not well arranged. So, there have some concerns that should be addressed before consider to accept.
Major concerns:
- Try to assign the stereochemistry of new compounds according biosynthetic pathway. For example, 4 was a biosynthetic precursor of 1/2, the stereochemistry of 4 can be tentatively assigned to be the same as 1/2. Similarly, the stereochemistry of 3, 5, and 6 can be assigned.
Response: We absolutely agree with the reviewer’s opinion in the elucidation of stereochemistry in the aspect of biosynthetic pathway. However, as shown in the stereochemistry for compounds 1 and 2 in the revised manuscript, they had different stereochemistry although they were isolated from the same strain. We checked the results of acetonide reaction and JBCA very carefully for compounds 1 and 2. As we indicated in the manuscript, the C-14, C-15, and C-16 in compound 1 have S, S, and R configurations, respectively. And four hydroxyl groups in the aliphatic chain of compound 2 are in “syn” orientation. Therefore, it is hard to say all the compounds isolated from this strain have the same stereochemistry.
In case of compounds 4, 5, and 7 with 1,2,4-triol system in the aliphatic chain, they were proposed to have the same stereochemistry as 1 based on the NMR shifts, coupling constants, and their synthetic origin. It was mentioned in the manuscript lines 238-240.
- The structure elucidation part need to be rearranged.
Response: As the reviewer’s suggestion, some parts of the structure elucidation were rearranged.
(1) All of the HMBC and COSY figures for new compounds should be presented.
Response: According to the reviewer’s comments, the HMBC and COSY figures of all the new compounds were presented in Figure 2.
(2) It’s better to write the structure elucidation for a new compound in a new paragraph.
Response: According to the reviewer’s suggestion, the description of the new structures started with a new paragraph in lines 126, 137, 148, and 152.
(3) The assignment of the long aliphatic chain is difficult to understand by interpretation of the 1H-1H COSY, because there are overlapping signals (CH2) in 1H NMR spectra. Therefore, the assignment of the long aliphatic chain needs additional supporting evidence.
Response: As the reviewer commented, there are many overlapping signals for CH2 in the 1H NMR spectra. However, the H2-10 in compound 1 appeared at δH 2.52 without being overlapped. By interpreting HMBC correlations from H2-10, signals for C-11 and C-12 could be assigned. Then signals for C-12 and C-13 could be assigned by interpreting HMBC correlations from H-14 at δH 3.42. H-14, H-15, and H-17 could be clearly assigned by 1H-1H COSY since they appeared separated from others. These key COSY and HMBC correlations were shown in Figure 2 and indicated in the manuscript (lines 85-88).
(4) Page 6, line 196, “The 13C resonances for the two methyl groups of the acetonide derivateve, …indicated a syn configuration of the dihydroxyl groups at C-14 and C-15 [30]”. The method and the reference for assigning the configuration of acetonide derivative are incorrect, please refer to literatures 10.1021/jo00103a012, 10.1074/jbc.RA120.015563.
Response: As the reviewer suggested, we cited one literature the reviewer recommended and another literature which shows the 13C-NMR and 1H-NMR resonances for 1,2-diol (Org. Lett. 2003, 5, 4231-4234).
Similarly, for page 7, line 232 “The 13C resonances for four methyl groups of the …” are incorrect. These configurations should be reassigned.
Response: As the reviewer commented, we carefully checked the 13C resonances for two acetonide groups. Four methyl carbons appeared at δC 30.5, 30.4, 20.23, and 20.17 and two acetonide ketal carbons appeared at δC 100.1, which indicated that both acetonide groups are in syn position based on the following references.
Reference: Acc. Chem. Res. 1998, 31, 9-17
Minor concerns
- Page 1 line 17, “Six new β-resorcylic acid derivatives (1–6 and 8)” should be “seven new β-resorcylic acid derivatives (1–6 and 8)”
Response: Since one compound was removed from the manuscript, “six” compounds are right.
- Table 1, Table 2 “δH, (J in Hz)”, should be “δH, (J in Hz)”, the data need to be rearranged accordingly.
Response: According to the reviewer’s suggestion, the data was rearranged in Table 1.
- “[M+Na-H2O]+” should be “[M + Na - H2O]+”, please check throughout the manuscript.
Response: It was revised throughout the manuscript.
- “syn configuration” should be “syn configuration”, please check all manuscript.
Response: According to the reviewer’s comments, “syn” was italicized throughout the manuscript.
- “JBCA” should be “JBCA”, please check all manuscript.
Response: The “J” in JBCA was italicized throughout the manuscript.
- OD values for 1 and 4 are too small. These two compounds were almost optically inactive. No need to list the OD values.
Response: According to the reviewer’s comments, the OD values for two compounds were removed.
- Page 15 line 421 showed the data of colletogloeic acid D diacetonide derivative (4a), which is not mentioned in the result.
Response: We truly apologize for the mistake. The diacetonide was obtained for compound 3 in the original manuscript (compound 2/2a in the revised manuscript).
- Page 4, line 127, please revise "1, 3, 4-triol" as "1, 2, 4-triol".
Response: It was revised in line 123.
- Page 7, line 238, please check the configuration at C-17 of compound 3 shown in Figure 1. It does not have a syn relationship with C-15.
Response: According to the reviewer’s comments, the structure for the compound was revised in the Figure 1.
- Page 4, lines 141-143, Although compound 5 has the same quasi molecular ion peak as 1 and 2, the molecular formula of compound 5 should not be the same as those of compounds 1 and 2 according to their structures shown in Figure 1. Please make a careful check of this point.
Response: According to the reviewer’s comments, we carefully checked the molecular ion peak and revised the manuscript.
Other concerns:
Are compounds 1 and 2 a pair of enol tautomers? Are compounds 1 and 2 stable? and how about compound 3 and 4?
Response: As we mentioned in the early part of this letter, structure of compound 2 was revised compound 4. In keto form, it is likely to form a macrolactone in the beta-resorcylic acid derivatives. All the isolated compounds in this study are stable.
Please contact me with any questions or concerns in regards to this submission by email at our conveniences.
Yours sincerely,
Sang Hee Shim, Ph.D./ Professor
Natural Products Research Institute, College of Pharmacy, Seoul National University
1 Gwanak-ro, Gwanak-gu, 08826 Seoul, South Korea
Tel: 82-2-880-2479
E-mail: sanghee_shim@snu.ac.kr

Reviewer 3 Report
Bang et al. described detailed isolation ad characterization for several new β
-resorcyclic acid derived natural products, especially compound 1 with rare enol moiety. The study also established stereochemistry for isolated compounds. The characterization is well discussed and data well presented.
However, the data interpretation needs to be improved for compound 2, as the NMR data listed in Table 1 and Figure S17 (also other supplementary information) are not consistent. I have highlighted some text and left comments in corresponding area (please see the attached file). It's recommended to double check the NMR spectra and structure of compound 2. Please also correct the corresponding result discussion:
- How many protons were in presence in the chemical shift range of 6.34-6.30 ppm?
- Where were the signals with chemical shifts 4.48 ppm and 3.82 ppm?
- Is there spectra evidence of C-9, with a chemical shift of 212.5 ppm?
Overall, this manuscript is recommended to be pushed, if sufficient correction has been made for above aspects.

Author Response
Thank you very much for your careful review on our manuscript. According to the comments, all comments were answered one by one as shown below. Thus, several parts of the revised manuscript have been corrected and indicated in red color concerning some modification in comparison with the previous manuscript.
In the revised manuscript, compound 2 in the original manuscript was removed because the structure of compound 2 was revised to compound 5 of the original manuscript (compound 4 of the revised manuscript) when writing the manuscript. Compound 2 should have been removed in original submission but it was not deleted by mistake. We truly apologize for causing confusion. Therefore, the numbers of the isolated compounds were revised accordingly as follows.
|
Compound No. of the submitted manuscript |
Compound No. of revised manuscript |
|
1 |
1 |
|
2 |
deleted |
|
3 |
2 |
|
4 |
3 |
|
5 |
4 |
|
6 |
5 |
|
7 |
6 |
|
8 |
7 |
|
9 |
8 |
|
10 |
9 |
Therefore, the description for compound 2 was totally removed from the manuscript.
Please contact me with any questions or concerns in regards to this submission by email at our conveniences.
Yours sincerely,
Sang Hee Shim, Ph.D./ Professor
Natural Products Research Institute, College of Pharmacy, Seoul National University
1 Gwanak-ro, Gwanak-gu, 08826 Seoul, South Korea
Tel: 82-2-880-2479

Round 2
Reviewer 2 Report
This reviewer is statified with the authors' revision and recommend to accept the manuscript as it is.
Author Response
Dear Editor,
Thank you very much for your careful review on our manuscript. According to the editor’s and reviewers’ comments, all comments were answered one by one as shown below.
Reviewer(s)' Comments to Author:
Reviewer #2:
This reviewer is satisfied with the authors' revision and recommend to accept the manuscript as it is.
Response: We appreciate for the kind review and comments.
Please contact me with any questions or concerns in regards to this submission by email at our conveniences.
Yours sincerely,
Sang Hee Shim, Ph.D./ Professor
Natural Products Research Institute, College of Pharmacy, Seoul National University
1 Gwanak-ro, Gwanak-gu, 08826 Seoul, South Korea
Tel: 82-2-880-2479
E-mail: sanghee_shim@snu.ac.kr

Reviewer 3 Report
Thanks very much for providing the corrections! It's a reasonable decision to delete the compound without correct characterization. Overall, the current manuscript looks significantly improved, which is recommended to be published.
However, there are still some errors in the manuscript and the supplementary data, please correct:
- Supplementary data for compound 3, is actually for compound 2. Same errors apply to supplementary data for compound 4, 5, 6, 8.
- And, what is the supplementary data for compound 2? Is this for the deleted compound? If so, please remove it. Please double check the numbering of the compounds for the manuscript and supplementary data carefully.
Author Response
Dear Editor,
Thank you very much for your careful review on our manuscript. According to the editor’s and reviewers’ comments, all comments were answered one by one as shown below.
Reviewer(s)' Comments to Author:
Reviewer #3:
Thanks very much for providing the corrections! It's a reasonable decision to delete the compound without correct characterization. Overall, the current manuscript looks significantly improved, which is recommended to be published.
However, there are still some errors in the manuscript and the supplementary data, please correct:
- Supplementary data for compound 3, is actually for compound 2. Same errors apply to supplementary data for compound 4, 5, 6, 8.
- And, what is the supplementary data for compound 2? Is this for the deleted compound? If so, please remove it. Please double check the numbering of the compounds for the manuscript and supplementary data carefully.
Response: According to the reviewer’s comments, we carefully checked the numbering of the compounds in the manuscript and the supplementary data.
As we commented before, compound 2 in the originally submitted manuscript was removed and the numbers of the compounds were rearranged accordingly. We removed the supplementary data for compound 2 and double checked the numbering of the compounds in the manuscript and supplementary data as attached.
Please contact me with any questions or concerns in regards to this submission by email at our conveniences.
Yours sincerely,
Sang Hee Shim, Ph.D./ Professor
Natural Products Research Institute, College of Pharmacy, Seoul National University
1 Gwanak-ro, Gwanak-gu, 08826 Seoul, South Korea
Tel: 82-2-880-2479
E-mail: sanghee_shim@snu.ac.kr
